# Time-trend in excess weight in Brazilian adults: A systematic review and meta-analysis

**Katia Kodaira**[1], **Flavia Casale Abe**[1], **Tais Freire Galvão**[2], **Marcus Tolentino Silva**[1]*

**1** Program of Pharmaceutical Sciences, University of Sorocaba, Sorocaba, São Paulo, Brazil, **2** Faculty of Pharmaceutical Science, Cidade Universitária Zeferino Vaz, State University of Campinas, Campinas, São Paulo, Brazil

☯ These authors contributed equally to this work.
* marcusts@gmail.com

**Data Availability Statement:** All relevant data are within the manuscript and its Supporting information files.

**Funding:** This study was financed in part by the Coordenação de Aperfeiçoamento de Pessoal de

## Abstract

### Background

This review aimed to estimate the time-trend prevalence of excess weight, overweight and obesity in the Brazilian adult population, from the 1970s–2020, through systematic review and meta-analysis (Protocol: CRD42018091002).

### Methods

A search for articles was conducted in the databases MEDLINE, EMBASE, Scopus, and LILACS up to June 2021. Studies that assessed excess weight, overweight and obesity in the adult population were eligible. Two authors selected studies, collected data and assessed the methodological quality of the studies. The primary outcomes were the prevalence of excess weight, overweight, and obesity by sex and period of years. Pooled prevalence and 95% confidence intervals (CIs) were calculated in the meta-analysis of the random effects model. Heterogeneity ($I^2$) was investigated by meta-regression and publication bias was investigated by Egger's test.

### Results

A total of 7,938 references were identified in the search strategies, of which eighty-nine studies and nine national surveys, conducted from 1974–2020, were included in the meta-analysis. The pooled prevalence of excess weight in Brazilian adults increased from 33.5% (95% CI: 25.0; 42.6%) in 1974–1990 to 52.5% (95% CI: 47.6; 57.3%) in 2011–2020. The pooled prevalence of overweight in Brazilian adults was 24.6% (95% CI: 18.8; 31.0%) from 1974–1990 and 40.5% (95% CI: 37.0; 43.9%) from 2011–2020. The pooled prevalence of obesity in Brazilian adults increased by 15.0% from 1974–1990 to 2011–2020. The increases were observed for both men and women in almost all periods. The prevalence of excess weight and obesity remained higher among women in all periods.

Nível Superior – Brasil (CAPES) – Finance Code 001. The funders had no role in study design, data collection and analysis, decision to publish, or preparation of the manuscript.

**Competing interests:** The authors have declared that no competing interests exist.

## Conclusions

A continuous increase in the prevalence of excess weight, overweight and obesity were observed over the years. The prevalence of excess weight affected half of Brazilian adults in the period from 2011–2020 and both sexes.

## Introduction

Obesity is a consequence of excessive fat accumulation in adipose tissue, and it is harmful to an individual's health [1]. The prevalence of this condition is on the rise and has become a global risk factor, affecting almost all countries, populations, and social classes [2]. Its evolution involves multiple factors, such as changes in lifestyle and food consumption, as well as socioenvironmental influences [3].

Obesity is also an important risk factor for major chronic noncommunicable diseases, such as diabetes, high blood pressure, cardiovascular diseases, cancer, and osteoarthritis [4–6]. The consequences of being overweight affect the quality of life and, development of psychosocial disorders and decrease life expectancy [7].

Brazil is one of the few middle-income countries that conducts frequent national cross-sectional, population-based surveys, either by the Brazilian Institute of Geography and Statistics or by educational research institutions. Comparative results of some national surveys have shown an increase in the prevalence of excess weight in adults, an important process in the nutritional and socioeconomic transition [8,9].

The first national survey in Brazil with information on the population's weight and height was the National Survey on Household Expenses, carried out in 1974, and the prevalence of excess weight was 21.4% in the adult population [10,11]. According to a national telephone survey carried out in Brazilian state capitals in 2019, the prevalence of excess weight was 75.7% [12]. The problem of excess weight is not exclusive to the adult population, as the prevalence in children and adolescents is also increasing [13,14].

Despite numerous national surveys and local studies, carried out in different Brazilian cities and regions, studies on the prevalence of excess weight, overweight and obesity in Brazilian adults are still fragmented. Previous analyses included nationally representative studies, telephone surveys or few regional studies, and to the best of our knowledge, there are no previous systematic reviews that synthesized available data [11,15–17]. Thus, we observed a need to group studies on the prevalence of excess weight and to assess the influence of study year on the estimates. This review aimed to estimate the time-trend prevalence of excess weight, overweight and obesity in the Brazilian adult population, from the 1970s–2020.

## Materials and methods

### Protocol and registration

A systematic review and a meta-analysis of cross-sectional studies were carried out. The study protocol was registered at the International Prospective Register of Systematic Reviews (PROSPERO) (Protocol: CRD42018091002).

### Eligibility criteria

Studies were considered eligible if they met the following criteria: a) cross-sectional, population-based studies with random sampling carried out in Brazil (city, regional or national),

which assessed the prevalence of excess weight, overweight, and/or obesity; b) studies including adults aged ≥18 (according to the age of majority criterion in Brazil) [18], of both sexes; c) studies regardless of the presence of comorbidities; d) longitudinal population-based studies that described the baseline of the study and not the segment; and e) studies without restrictions on year, publication status, or language. Studies carried out in groups of specific populations, such as indigenous people, pregnant women, institutionalized persons, and children were excluded.

## Search strategy

The search for articles was carried out in the databases MEDLINE (via PubMed), EMBASE, Scopus, and LILACS. As a strategy to identify potentially eligible studies not indexed in databases, we screened the references of relevant publications. Also, other sources like letters, personal opinions, books, or conference abstracts were considered as a strategy to identify other possible additional studies. The search strategy is available in Supporting information S1 Table. Searches were held in March 2018 and updated in June 2021.

## Study selection and data collection

Study selection was performed using Covidence platform (www.covidence.org) [19]. This Platform allows to remove duplications and the independent selection of titles, abstracts, and full texts, as well as their extraction. After removing duplicate articles, two independent authors (KK, FCA) selected the articles by title and abstract, according to the eligibility criteria. Any mention of excess weight, overweight or obesity was taken in consideration to include the study in the full text review.

Relevant articles were retrieved for full text review. Studies that used data from a same research with similar results were assessed, and the publication that included the data and results in a more complete and detailed way was included, and the others were excluded. Selected articles that referred to national surveys were replaced by the original microdata, when available. Disagreements at all stages were solved by consensus or decided by a third author (MTS).

The following data were extracted by two authors from the selected studies, using standardized spreadsheet: author; year; place (city or state); macro-region of the country; total sample size; women sample size; prevalence of excess weight, overweight, and obesity by sex; overall prevalence of excess weight, overweight, and obesity; age group; self-reported or measured data; body mass index (BMI) classification. The BMI categories considered were excess weight (BMI ≥ 25kg/m$^2$), overweight (25kg/m$^2$ ≤ BMI < 30 kg/m$^2$), and obesity (BMI ≥ 30kg/m$^2$). These categories were defined according to World Health Organization (WHO) guidelines on BMI classification [20]. Studies that considered other BMI classifications were classified into the WHO classification.

## Quality assessment of included studies

Two authors (KK, FCA) independently assessed the methodological quality of the studies using a standardized tool [21], consisting of nine items: i) was the sample frame appropriate to address the target population?, ii) were study participants sampled in an appropriate way?, iii) was the sample size adequate?, iv) were the study subjects and the setting described in detail?, v) was the data analysis conducted with sufficient coverage of the identified sample?, vi) were valid methods used for the identification of the condition?, vii) was the condition measured in a standard, reliable way for all participants?, viii) was there appropriate statistical analysis?,

and ix) was the response rate adequate, and if not, was the low response rate managed appropriately?.

Disagreements were resolved by the third author (MTS). For each study, the value 0 corresponded to a limitation in the respective item and the value 1, corresponded to a met expected criterion. The total score could range from zero (low quality) to nine (high quality).

### Data synthesis and statistical analyses

Data analyses were performed using Stata® version 14.2 (Stata Corp, College Station, Tx, United States). The primary outcomes were the prevalence of excess weight, overweight, and obesity with a 95% confidence interval (CI) by sex and year of study. The years considered were: 1974–1990 (due to few studies in the period), 1991–2000, 2001–2010, and 2011–2020.

The results were pooled by meta-analysis using the random effects model described by Der-Simonian and Laird and the double-arcsine transformation for variance stabilization as proposed by Freeman-Tukey [22]. Heterogeneity was assessed by the chi-square test with a significance of $p < 0.10$, and its magnitude was determined by the I-square ($I^2$) [23]. To identify the possible causes of heterogeneity, a meta-regression was performed using the Knapp-Hartung method [24]. The effects of the prevalence of excess weight, overweight, and obesity; year of data collection; and methodological quality of the studies were evaluated. Potential publication bias was assessed using a funnel plot and Egger's test [25].

### Results

The literature search yielded a total of 7,938 potentially eligible articles (Fig 1). Based on titles and abstracts, we selected 236 articles for full text review. Eighty-nine studies carried out between 1974 and 2020 met the eligibility criteria and were included in the research. No new studies were identified by searching the references of the included articles. Microdata from nine national surveys were included. The references of all 85 reports from 89 included studies are listed in Supporting Information S2 Table.

Some studies used self-reported data to calculate BMI. Most studies scored between 7 and 9 in the evaluation of methodological quality. No study was excluded because of its low methodological quality (S2 Table).

The pooled prevalence of excess weight in Brazilian adults increased from 33.5% (95% CI: 25.0; 42.6%) in the 1974–990 to 52.5% (95% CI: 47.6; 57.3%) in the 2011–2020 period (Table 1). In both sexes, the prevalence of excess weight increased in all periods and the prevalence remained higher among women.

The pooled prevalence of overweight in Brazilian adults was 24.6% (95% CI: 18.8; 31.0%) in the 1974–1990, and it has increased over the years. In the last decade, the prevalence of overweight was 40.5% (95% CI: 37.0; 43.9%). In almost all periods, there was a predominance of overweight men compared to women, except for the period from 1974–1990, in which the prevalence of overweight was 26.0% (95% CI: 20.4; 32.0%) for women and 22.8% (95% CI: 16.7; 29.6%) for men (Table 1).

The pooled prevalence of obesity in Brazilian adults increased by 15.0% from 1974–1990 to 2011–2020. Increases were observed for both men and women, although the increase in the prevalence of obesity was greater among women than men over the past three and a half decades analyzed in this study.

Trends in the prevalence of excess weight, overweight and obesity in Brazilian adults have increased over the years, both for men and women (Fig 2). All trends were statistically significant (p<0.001).

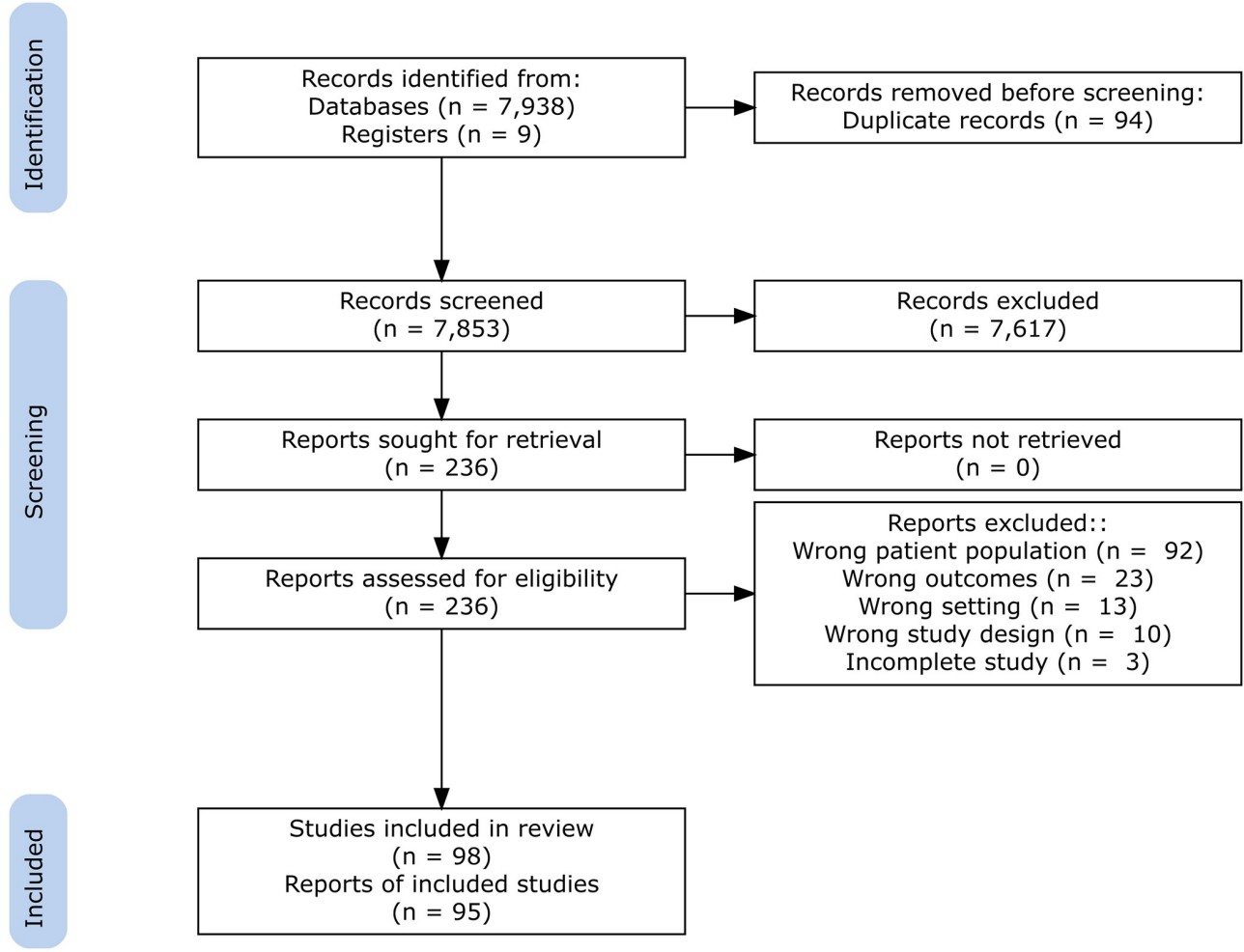

**Fig 1. PRISMA flow chart showing the selection of studies included in systematic review.**

All estimates were heterogeneous. Subgroup analysis did not identify causes of heterogeneity. The excess weight prevalence funnel plot showed high asymmetry (S1 Fig). and potential publication bias, which was confirmed by the presence of the effect of small studies (p = 0.001). This behavior was also observed for the overweight and obesity outcomes (S2 and S3 Figs).

## Discussion

### Summary of results

The prevalence of excess weight, overweight and obesity in Brazilian adults has been increasing since the mid-1970s. This increase is reflected in both sexes and in all periods of years analyzed in the review (1974–2020).

### Research validity and limitations

The included studies adopted probabilistic sampling with different strategies, ensuring greater representativeness of the sampled population. The research covered a wide period, between

**Table 1. Meta-analysis of excess weight, overweight and obesity in Brazilian adults, by sex and period of years, from 1974 until 2020.**

| Period | 1974–1990 | | | | 1991–2000 | | | | 2001–2010 | | | | 2011–2020 | | | |
|---|---|---|---|---|---|---|---|---|---|---|---|---|---|---|---|---|
| | Prev. (%) (95% CI) | $I^2$(%) | studies | Sample size | Prev. (%) (95% CI) | $I^2$(%) | Studies | Sample size | Prev. (%) (95% CI) | $I^2$(%) | Studies | Sample size | Prev. (%) (95% CI) | $I^2$(%) | Studies | Sample size |
| Excess weight | 33.5 (25.0; 42.6) | 99.9 | 4 | 164,425 | 40.0 (35.8; 44.3) | 98.9 | 19 | 47,286 | 47.7 (45.1; 50.2) | 99.5 | 50 | 331,003 | 52.5 (47.6; 57.3) | 99.3 | 25 | 68,542 |
| Men | 37.5 (28.5; 46.9) | 99.7 | 4 | 80,851 | 33.7 (25.3; 42.7) | 98.8 | 11 | 11,459 | 44.5 (40.6; 48.5) | 99.3 | 28 | 128,078 | 49.1 (43.4; 54.8) | 98.1 | 12 | 22,994 |
| Women | 28.5 (20.5; 37.3) | 99.7 | 4 | 83,574 | 41.8 (37.0; 46.6) | 98.0 | 14 | 24,843 | 45.2 (42.4; 48.0) | 99.1 | 33 | 169,756 | 62.9 (61.2; 64.5) | 98.9 | 14 | 31,175 |
| Overweight | 24.6 (18.8; 31.0) | 99.7 | 4 | 164,425 | 31.8 (28.8; 35.0) | 97.6 | 14 | 41,384 | 33.8 (32.1; 35.4) | 98.6 | 41 | 299,300 | 40.5 (37.0; 43.9) | 98.1 | 14 | 57,216 |
| Men | 22.8 (16.7; 29.6) | 99.5 | 4 | 80,851 | 33.0 (27.8; 38.3) | 95.6 | 7 | 9,258 | 37.2 (34.7; 39.6) | 97.7 | 20 | 115,426 | 43.7 (39.8; 47.6) | 95.7 | 9 | 22,260 |
| Women | 26.0 (20.4; 32.0) | 99.4 | 4 | 83,574 | 29.7 (27.4; 32.2) | 90.8 | 9 | 21,142 | 33.9 (31.7; 36.1) | 98.3 | 24 | 151,765 | 42.6 (37.7; 47.5) | 98.0 | 10 | 29,312 |
| Obesity | 8.6 (5.7; 11.9) | 99.6 | 4 | 164,425 | 16.9 (14.0; 20.2) | 98.5 | 16 | 42,141 | 17.2 (15.6; 19.0) | 99.2 | 39 | 297,634 | 23.9 (21.8; 26.0) | 96.6 | 17 | 60,675 |
| Men | 5.3 (3.3; 7.8) | 98.9 | 4 | 80,851 | 12.7 (8.8; 17.0) | 97.4 | 11 | 11,459 | 14.6 (12.3; 17.1) | 98.8 | 20 | 115,256 | 20.4 (18.4; 22.4) | 87.4 | 8 | 21,788 |
| Women | 11.2 (7.7; 15.3) | 99.3 | 4 | 83,574 | 20.9 (16.4; 25.8) | 98.6 | 13 | 24,041 | 19.3 (17.3; 21.4) | 98.7 | 24 | 151,712 | 29.7 (26.3; 33.2) | 96.5 | 10 | 29,726 |

Prev.(%), prevalence. 95%CI, 95% confidence interval. $I^2$, heterogeneity.

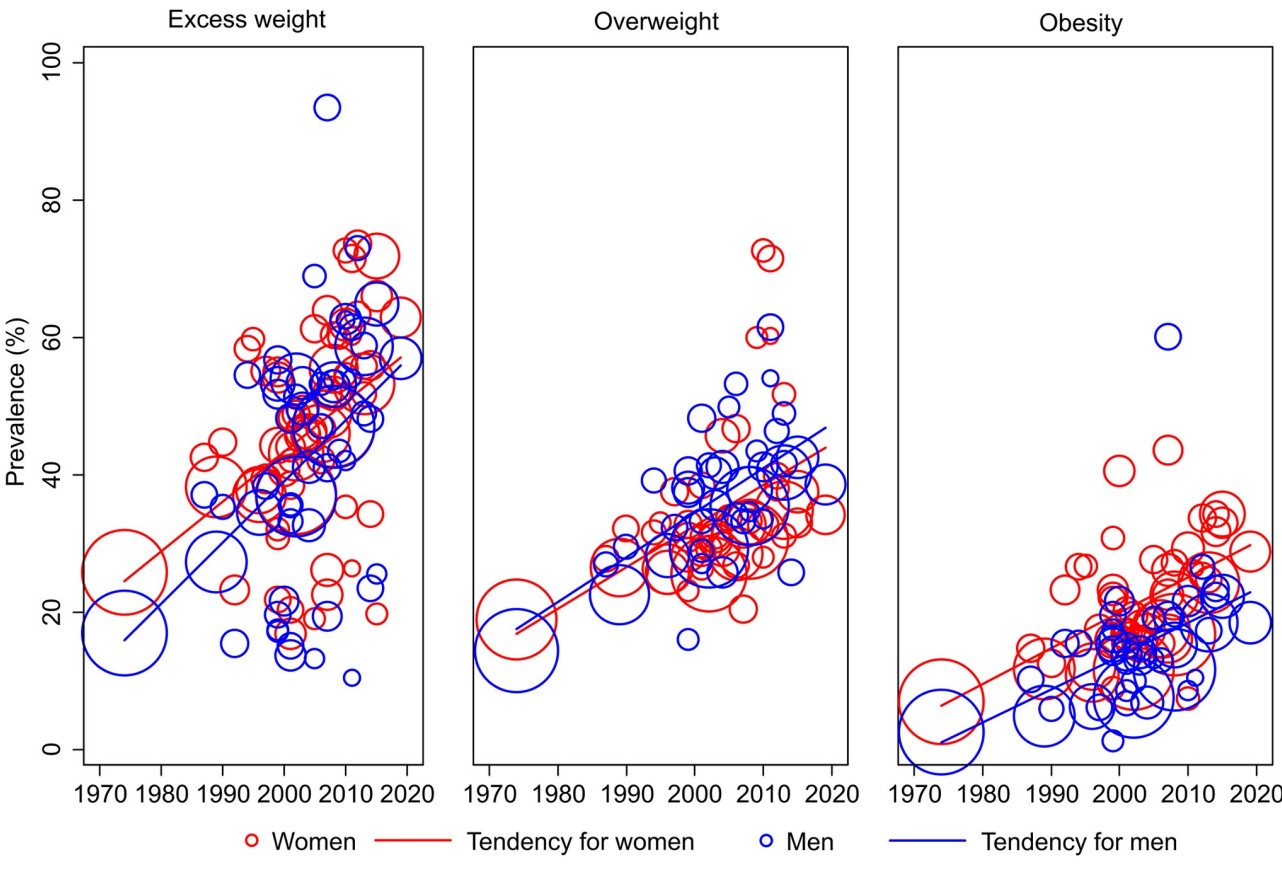

**Fig 2. Tendency of excess weight, overweight and obesity in Brazilian adults, by sex and year, from 1974 to 2020.**

the mid-1970s and 2020. The data presented were collected during different phases of an increasing trend in the prevalence of overweight and obesity, ensuring the temporality of results. The amplitude of the periods covered by the review made it possible to assess the growing trend.

Studies were not grouped by sample size, urban or rural location, age group, or socioeconomic status, which are factors that can affect the prevalence of excess weight and heterogeneity [26–28]. We also observed a trend to publish results with high prevalence. Our hypothesis is that studies with low prevalence were omitted because of a lack of interest in the scientific environment or even by the author's decision not to publish the work, which fails to reflect reality [29].

Most studies used the WHO [20] as a reference to classify overweight and obesity. Some studies involving the elderly population adopted the Lipschitz [30] or the Pan American Health Organization [31] classifications, which consider the physical changes in weight and height during the aging process. Ideally, the adoption of the same cutoff point to define the outcome would bring greater validity [1]. The use of self-reported data in certain publications may also affect the results found. The variability in self-reported weight and height values induces changes in BMI and, therefore, in the prevalence of excess weight [32,33]. Anthropometric measures in studies based on self-reported data have shown a trend to underestimate weight and overestimate height [34,35]. Women tend to report lower weight values and men tend to report higher height values [32].

The prevalence of excess weight may also be influenced by survival bias. The mortality rate of individuals with moderate obesity (BMI from 30.0–39.9 kg/m$^2$) can reduce life expectancy by up to four years, and in cases of severe obesity (above 40 kg/m$^2$), the decrease can reach ten years [36]. It is possible that obese people are missed by random sampling at the time of population surveys. Survival bias changes the lifetime prevalence, but it does not change the prevalence that we estimate.

### Comparison with the relevant literature and interpretation of the findings

The prevalence of excess weight has been increasing in Brazil. Similar results from other global studies show the trend of increasing prevalence of excess weight and obesity, both in high income, low-middle and low- income countries [4,17,37].

This review has found a progressive growth of excess weight in Brazil over the years. The United States, a country with one of the highest rates of obesity in the world, presented a prevalence of obesity of 14.6% in the 1970s, with an increase to 38.7% in 2016 [38,39]. In a study that evaluated different countries in Europe in 2014, the prevalence of overall excess weight was 53.1%, close to the Brazilian reality [40]. A Chinese study also observed increasing trends of overweight and obesity even considering different BMI cutoff points for Asian countries [41], as recommended by the WHO. The prevalence of overall obesity was 2.2% in 1989 and 14.0% in 2011.

We found a higher prevalence of excess weight and obesity in women than in men in this review. A worldwide study showed an increasing prevalence of obesity among women of all age groups, regardless of the country's economic development [4]. Another review covering women from developing countries found an increase in obesity, mainly in Latin America and the Caribbean, the Middle East, North Africa, and Central Europe [42]. A Chinese study assessed the prevalence of excess weight among women and men between 1991 and 2000 [43]. The prevalence of excess weight was 14.5% for women and 9.6% for men in 1991. In 2000, there was an increase for both sexes but mainly among women, in which the prevalence was 26.5%, with 20.0% for men. Compared with our results, the prevalence of excess weight in Chinese population was lower than that found for both women and men within the same period (1991–2000).

An Ethiopian study investigated trends in excess weight in women from 2000–2016. The observed prevalence of excess weight was much lower for Ethiopian women than that for women in Brazil. The prevalence of excess weight in Ethiopian women was 10.9% in 2000, 14.9% in 2011, and 21.4% in 2016 [44].

### Conclusion

The prevalence of excess weight in Brazil is rising. An alert must be raised to the problem and the urgent need for initiatives aimed at controlling overweight and obesity, since excess weight affected half of Brazilian population. Policy-makers and health managers should reassess current interventions and policies to prevent a serious obesity epidemic and initiate effective preventive measures to reduce excess weight in the country.

### Supporting information

**S1 Fig. Funnel plot of excess weight prevalence in Brazilian adults, from 1974 until 2020.** (DOCX)

**S2 Fig. Funnel plot of overweight prevalence in Brazilian adults, from 1974 until 2020.** (DOCX)

**S3 Fig. Funnel plot of obesity prevalence in Brazil, from 1974 until 2020.**
(DOCX)

**S1 Table. Electronic search strategy.**
(DOCX)

**S2 Table. Main characteristics and quality assessment of included studies in meta-analysis.**
This is the S2 Table legend. M, measured. SR, self-reported. BMI, body mass index. BR, Brazil. WHO, World Health Organization. PAHO, Pan American Health Organization. ENDEF, Estudo Nacional de Despesa Familiar (National Survey on Household Expenses). PNSN, Pesquisa Nacional de Saúde e Nutrição (National Survey on Health and Nutrition). PNDS, Pesquisa Nacional de Demografia e Saúde (National Demography and Health Survey). PPV, Pesquisa sobre Padrões de Vida (Living Standards Survey). POF, Pesquisa de Orçamentos Familiares (Household Budget Survey). PDSD, Pesquisa Dimensões Sociais das Desigualdades (Social Dimensions of Inequalities Survey). PNS, Pesquisa Nacional de Saúde (National Health Survey). ELSI, Estudo Longitudinal da Saúde dos Idosos (Brazilian Longitudinal Study of Aging). [a]WHO–overweight:BMI $\geq$ 25 and <30 kg/m2, obesity:BMI $\geq$ 30kg/m2; Lipschitz–underweight: BMI less than 22kg/m2, overweight: BMI more than 27kg/m2; PAHO–underweight: BMI $\leq$ 23 kg/m2, overweight: BMI $\geq$ 28 kg/m2 and < 30 kg/m2, obesity: BMI $\geq$ 30 kg/m2. [b]Critical appraisal according to The Joanna Briggs Institute Critical Appraisal Checklist for Studies Reporting Prevalence Data: 1. Was the sample frame appropriate to address the target population? 2. Were study participants sampled in an appropriate way? 3. Was the sample size adequate? 4. Were the study subjects and the setting described in detail? 5. Was the data analysis conducted with sufficient coverage of the identified sample? 6. Were valid methods used for the identification of the condition? 7. Was the condition measured in a standard, reliable way for all participants? 8. Was there appropriate statistical analysis? 9. Was the response rate adequate, and if not, was the low response rate managed appropriately?.
(DOCX)

## Author Contributions

**Conceptualization:** Katia Kodaira, Flavia Casale Abe, Marcus Tolentino Silva.

**Data curation:** Katia Kodaira, Marcus Tolentino Silva.

**Formal analysis:** Katia Kodaira, Flavia Casale Abe, Tais Freire Galvão, Marcus Tolentino Silva.

**Investigation:** Katia Kodaira, Flavia Casale Abe, Marcus Tolentino Silva.

**Methodology:** Katia Kodaira, Flavia Casale Abe, Tais Freire Galvão, Marcus Tolentino Silva.

**Project administration:** Katia Kodaira, Marcus Tolentino Silva.

**Software:** Marcus Tolentino Silva.

**Supervision:** Tais Freire Galvão, Marcus Tolentino Silva.

**Validation:** Katia Kodaira, Flavia Casale Abe, Tais Freire Galvão, Marcus Tolentino Silva.

**Visualization:** Katia Kodaira, Flavia Casale Abe, Tais Freire Galvão, Marcus Tolentino Silva.

**Writing – original draft:** Katia Kodaira, Marcus Tolentino Silva.

**Writing – review & editing:** Katia Kodaira, Flavia Casale Abe, Tais Freire Galvão, Marcus Tolentino Silva.

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
