## [Decision Letter · Decision Letter 0]

13 May 2021

PONE-D-20-38212

Time-trend of excess weight in Brazil: a systematic review and meta-analysis

PLOS ONE

Dear Dr. Silva,

Thank you for submitting your manuscript to PLOS ONE. After careful consideration, we feel that it has merit but does not fully meet PLOS ONE’s publication criteria as it currently stands. Therefore, we invite you to submit a revised version of the manuscript that addresses the points raised during the review process.

Many suggestions were made and some important issues need to be addressed that include originality of the work, the classification of participants as being adolescents, justifying the methodology employed, discrepancies in the data between results and figures. I addition present the data yearly instead of an average and separated meta-analyses. Additional references are needed to substantiate some statements and a more comprehensive PubMed search for similar studies.

Please address all recommendations and questions raised by the 3 reviewers and answer them completely. We look forward to the revised manuscript.

We look forward to receiving your revised manuscript.

Kind regards,

Cesario Bianchi

Academic Editor

PLOS ONE

Additional Editor Comments:

Dear Dr. Silva:

Thank you for submitting your interesting manuscript that was reviewed by 3 experts.

Many suggestions were made and some important issues need to be addressed that include originality of the work, the classification of participants as being adolescents, justifying the methodology employed, discrepancies in the data between results and figures. I addition present the data yearly instead of an average and separated meta-analyses. Additional references are needed to substantiate some statements and a more comprehensive PubMed search for similar studies.

Please address all recommendations and questions raised by the 3 reviewers and answer them completely. We look forward to the revised manuscript.

Journal Requirements:

"No"

Reviewers' comments:

Reviewer's Responses to Questions

**Comments to the Author**

1. Is the manuscript technically sound, and do the data support the conclusions?

Reviewer #1: Yes

Reviewer #2: Partly

Reviewer #3: Yes

2. Has the statistical analysis been performed appropriately and rigorously? 

Reviewer #1: Yes

Reviewer #2: Yes

Reviewer #3: I Don't Know

3. Have the authors made all data underlying the findings in their manuscript fully available?

Reviewer #1: Yes

Reviewer #2: Yes

Reviewer #3: Yes

4. Is the manuscript presented in an intelligible fashion and written in standard English?

Reviewer #1: Yes

Reviewer #2: Yes

Reviewer #3: Yes

5. Review Comments to the Author

Reviewer #1: I appreciate the opportunity to review this manuscript. The manuscript "Time-trend of excess weight in Brazil: a systematic review and meta-analysis" summarizes studies of prevalence of the excess of weight and assesses the temporal tendency of this indicator in the adult Brazilian population. This is an important area of research and strengthens the formulation of public policies, plans and actions to reduce, prevent and treat the excess of weight in Brazil. The authors provided all the data extracted from the studies included in the systematic review and meta-analysis and the PRISMA-P checklist. The manuscript brings interesting results, however, there are details that need to be clarified or explained more specifically for the final publication (see the specific comments below).

Title:

• I suggest including the studied population in the title.

Introduction:

• I suggest bringing evidence-based studies which prove that there is no previous systematic review about the topic addressed. This information cannot be empirical;

• Including epidemiological data about the excess of weight in Brazil;

• Evaluating the possibility of including in the “study objective” the period of time in which the temporal tendency was assessed.

Methodology

• Were the researchers who assessed the eligibility criteria blindfolded? It is important to describe this;

• Why was the adult population aged 18 or older considered? According to WHO criteria, for the assessment of nutritional status/growth curves, people aged between 10 and 19 years old are considered adolescents. I recommend justifying this choice of the methodology;

• The article is registered in PROSPERO, but it is not up to date. I recommend reviewing it.

Reviewer #2: Manuscript Review

PONE D-20-38212-R1

April 17, 2021.

Title: “Time-trend of excess weight in Brazil: a systematic review and meta-analysis.”

Summary

The authors conducted a systematic review and meta-analysis involving the adult Brazilian population up to December 2020. They investigated the time trend in excess weight in Brazil and found an overall prevalence of 46.7%, with a 12% increase between 2005 and 2016.

According to the authors, excess weight affected half of the adult Brazilian population of both sexes and in all regions of Brazil.

Study strengths

• Meta-analysis.

• The wide period investigated.

Main study limitations

• The authors did not consider other important keywords in the search strategy, such as “excess weight” and “excess body weight”, a major limitation.

Points that need clarification

Study overview

The field of the study is important and relevant since excess weight is a serious public health problem worldwide. However, the study has several methodological problems mainly in the Methods and Results sections. The justification for the study needs to be included in the introduction

Title:

1. The title should state that the study was carried out in adults: “Time trend in excess weight in Brazilian adults: a systematic review and meta-analysis.”

Abstract:

2. At the end of the conclusion, the authors state: “Excess weight affected half of the adult Brazilian population, both sexes and in all regions of Brazil.”. However, they do not report the most prevalent sex and region in the Results section.

3. The authors reported that 86 studies and 8 national surveys were included in the study. However, the results are different from in Figure 1 “…(n = 91), that refers to 94 studies”?

Introduction:

4. Page 3, line 59: Please add references.

5. Page 3, lines 64-65: “This review aimed to estimate the time-trend prevalence of excess weight in the Brazilian adult population.”. Please include the period investigated.

6. I was unable to locate the justification for the study. Why did the authors decide to do a systematic review on excess weight involving Brazilian adults?

Methods:

7. The Methods section is poorly described. The authors should follow the PRISMA checklist to better describe the methods of the study.

8. Page 4, line 71-73: Did the authors include a grey literature search? The grey literature refers to unpublished research or published in non-commercial form and allows a more comprehensive search strategy.

9. Page 4, line 75: Did the authors consider longitudinal studies in the eligibility criteria? If so, it is important to mention. Otherwise justify. Also, were letters, personal opinions, book chapters or conference abstracts considered?

10. Page 4, line 78: Were studies involving people with other comorbidities (diabetes, cardiovascular diseases, …) also excluded? This must be mentioned.

11. Page 4, line 80: “…and children (< 18 years old)…” repeated from “b”, line 77.

12. Page 5, line 95: Include checklist reference. I suggest inserting the checklist as complementary material.

13. Page 5, lines 99-101: “The total score obtained could range from zero (low quality) to nine (high quality). Surveys were considered adequate when they scored ≥ 6.” Please justify or include a reference explaining why the authors chose scored ≥ 6 as adequate.

14. S1 Table. Electronic search strategy: It is difficult to understand why the authors did not use other important key words in the search strategy, such as “excess weight” and “excess body weight”. In fact, the keyword “excess weight” appears in the title of this review! In a quick search using the terms “(excess weight OR excess body weight) AND (brazil OR brasil) AND (prevalence OR prevalences)”, I found 642 articles!

15. Still regarding the search strategy, many studies report the prevalence of overweight/obesity/excess weight throughout the article and not in the title or abstract because it is not the main objective of the article. Did the authors consider this fact in the search?

Results:

16. The authors should follow the PRISMA checklist to better describe the results of the study.

17. Fig 1: What do the authors mean by “Additional records through other sources”? This explanation was not mentioned in the Methods section.

18. Page 6, line 124, S2 Table: I suggest including S2 Table as Table 1 in the Results section. Change the current Table 1 to Table 2. Qualitative data must be shown before meta-analysis data.

19. Page 6, line 129: “The prevalence …” of what? Excess weight? Overweigh? Please, define. The same inconsistency in line 131.

20. Page 6, lines 125, 127, 131-32, Table 1, and Fig 2: Which means, “excess weight”? The authors did not define this term before and did not consider it in the search strategy.

21. Page 6, lines 132-34: “Between 1974 and 2004, the prevalence of excess weight was 40.2% (95% CI: 36.8; 43.7%) and from 2005 onwards, it has been 51.7% (95% CI: 134 48.8; 54.6%).” Does the prevalence reported by the authors refer to an increase? Was there an increase of 40.2% and 51.7% over the periods mentioned?

22. Table 1: Please include “n” for each category.

23. Figure 1, title: Again, update the word “excess weight”.

24. S1 Figure was not cited in the Results section. Also, include the definition of “ES” in the S1, S2 and S3 Figures.

25. Page 6, line 144: Use only one type of denomination (sex: male/female or gender: men/women) throughout the manuscript, including tables and figures.

26. Page 8, lines 143-46: “In almost all regions and in national surveys, except in North region, there was a predominance of male overweight compared to female”. This information is from Table 1, not S2 Fig. Add Table 1 at the end of the paragraph.

Discussion:

27. Page 9, line 172: “We also observed a trend to publish results with high prevalence”. I wouldn’t say that! Is that a phenomenon that occurs only in Brazil? However, the authors need to better support this statement with more studies conducted in Brazil and in other countries.

28. Page 9, line 185: “The results…”. Which results?

29. Page 10, line 191: “The prevalence of excess weight has been increasing.” In Brazil? Worldwide? Please mention!

30. Page 10, line 192: “…trend of increasing prevalence, …”. Which prevalence? Excess weight? Overweight? Obesity?

31. Page 10, line 197: “…in 2016; high values …” high or higher values?

32. Page 10, lines 197-98: “…In a study that evaluated different countries in Europe, the prevalence of overall excess weight was 53.1% and obesity, 15.9%, close to the Brazilian reality”. Were these studies carried out over the same period?

Reviewer #3: Reviewer Feedback on PONE-D-20-38212

This is a well-written and well-conducted systematic review on an important topic, and I recommend it for publication. There are, however, a number of issues which should be addressed beforehand.

Of note, I’m not a statistician or a mathematician. While the statistical analyses done in the study seem sound to me, their validity should also be assessed by a meta-analysis specialist.

General issues

You seem to have lumped all data from 1974 to 2016 together into one meta-analysis, which yielded average prevalence data for this time period. I’m not sure how meaningfull it is to combine data from such a long time period, in particular given the enormous changes Brazil has undergone during this time period. For decicion-makers in policy and practice and researchers it would be more useful to know how high the prevalence was in 1974, and how high it was in 2016, rather than having an average figure for the whole time period. Please explain, and consider doing separate meta-analyses for different time periods or points of time.

Minor issues

Abstract, line 34-35 (“The overall prevalence of excess weight in Brazilian adult population was 46.7% (95% CI: 43.6; 49.7%).”): Please specify to which time point or time period this refers.

Introduction, line 51-52 (“The consequences of being overweight… decrease life expectancy (7) due to an increased risk of premature death (1,8).“): Isn’t a decrease in life expectancy just another way of saying that the risk of premature death is increased? I find it somewhat strange to say that one is the consequence of the other.

Introduction, line 53-54 (“Brazil is one of the few developing countries that conducts frequent national cross-sectional, population-based surveys..“): Wouldn’t it be more appropriate to describe Brazil as a middle-income country?

Introduction, line 55-57 („Comparative results of some national surveys have shown an increase in the prevalence of excess weight in adults, an important event in the

process of nutritional and socioeconomic transition“): I would frame the increase in the prevalence of excess weight as a development or process rather than an event.

Introduction, line 57-59 (“This is not exclusive to the adult population, since there is a higher prevalence in children and adolescents, suggesting that the problem becomes even more serious over the years.“): Please provide a reference for the statement that the prevalence of obesity is hight among children and adolescents than among adults in Brazil (in most countries I am aware of it’s the other way round).

Introduction, line 60-62 („Despite the numerous national surveys carried out, studies on prevalence of excess weight, overweight and obesity with Brazilian adults still so far fragmented and to the best of our knowledge, there are no previous systematic review that synthesized available data.): The NCDRisC Risk Factor Collaboration has published multiple pooled analyses of body weight trajectories, which include Brazil and which are based on reviews of existing surveys, see https://ncdrisc.org/publications.html.

Methods, line 67-70 (Protocol and registration): Please state if this was an a priori protocol, i.e. a protocol developed before the study began, and if there were any deviations from the protocol (i.e. anything you ended up doing differently than envisioned in the protocol – which is fine as long as it is stated transparently and reasonably justified).

Methods, line 71-73: I would propose to insert a separate sub-heading for the section on search methods (line 71-73).

Eligibility criteria, line 76-77 („cross-sectional, population-based studies and random sampling, carried out in Brazil“): Please specify if studies had to be conducted on a national basis, i.e. if it was an eligibility criteria that they were representative for the whole of Brazil, or if you also included studies from specific regions within Brazil.

Eligibility criteria, line 76-77 („cross-sectional, population-based studies and random sampling, carried out in Brazil“): Please specify if you also included cohort studies.

Methods, line 95-97 (“The methodological quality of included studies was assessed using standardized tool The Joanna Briggs Institute Critical Appraisal Checklist for Studies Reporting Prevalence Data)“). As not all readers may be familiar with this tool, please consider providing a concise summary of it (2-3 sentences, and a table with the criteria used by the checklist).

Results, line 116-117 („No new studies were identified by searching references of the included articles.“): Searchign the reference lists of included studies is indeed recommended, and should also be mentioned in the methods section under the sub-heading „Search methods“.

Results, line 125 („The overall prevalence of excess weight was 46.7%“). Please specify to which year this statement refers. This also applies to all similar statements in the subsequent sections of the manuscript.

Results, line 131-13 („Trends on prevalence of excess weight has been increasing over the years (Fig 2).“): Figure 2 seems to show that the level of prevalence of obesity is increasing, which is not the same like saying that the trend on the prevalence of obesity is increasing (the latter seems to imply that the rate of chenge ist increasing.

Results, line 132-134 („Between 1974 and 2004, the prevalence of excess weight was 40.2% (95% CI: 36.8; 43.7%) and from 2005 onwards, it has been 51.7% (95% CI: 48.8; 54.6%).“): To me, it seems a bit strange to lump such a long time period (1974-2004) together, and to say that from 2005 onwards, the prevalence has been exactly 51.7%. In reality, the prevalence seems to have been constantly rising. Besides, why have you chosen this specific cut-off (i.e. the year 2004/2005)?

Table 1, as well as several instances throughout the manuscript: It seems as if you used the term „excess weight“ as summary category for both simple overweight (i.e. overweight without obesity) and obesity. Besides, you seem to use the category „overweight“ for simple overweight only (i.e. a BMI from 25 to 30), not including those with obesity. This should be made explicit somewhere.

Results, line 141 („The overall prevalence of overweight in Brazil was 34.3%“): Please specify to which time period this refers. The same applies to similar statements in the following sections. Whenever a prevalence rate is given, the time frame should be stated explicitly.

Results, line 154-156 (“The excess weight prevalence funnel plot showed a high asymmetry (S4 Fig.). and a potential publication bias, which was confirmed by the presence of the effect of small studies (p = 0.001).“): Please state in which direction the data is likely to be biased (i.e. did small studies find, on average, higher prevalence rates than bigger studies, or was it the other way round?).

Discussion, line 159 (“The prevalence of excess weight has been increasing since the mid-1970s.“): While it may be obvious, it may still be worthwhile stating that this sentence refers to Brazil.

Discussion, line 185-188 („The results may also be influenced by survival bias. (…)“). Survival bias implies that the lifetime prevalence of obesity may be higher than the estimates for the point prevalence of obesity which you derived. However, in my understanding it does not impact the validity of the the latter – it just implies that they are lower than the lifetime prevalence. Please consider making this clearer in the discussion.

Further issues: The PubMed search syntax which you used was rather simple, and may have missed relevant studies. Besides, you seem not to have done a citing-studies search (also known as forward-citation search), i.e. a search for studies that cited the studies included in your review (such searches can be done in databases such as Scopus). Please consider mentioning this in the limitations section.

6. PLOS authors have the option to publish the peer review history of their article (what does this mean?). If published, this will include your full peer review and any attached files.

Reviewer #1: **Yes: **Mariana Balestrin

Reviewer #2: No

Reviewer #3: **Yes: **Peter von Philipsborn

---

## [Author Response · Author response to Decision Letter 0]

26 Aug 2021

We attached a specific file with our response.

---

## [Decision Letter · Decision Letter 1]

10 Sep 2021

Time-trend in excess weight in Brazilian adults: a systematic review and meta-analysis

PONE-D-20-38212R1

Dear Dr. Silva,

We’re pleased to inform you that your manuscript has been judged scientifically suitable for publication and will be formally accepted for publication once it meets all outstanding technical requirements.

Kind regards,

Cesario Bianchi

Academic Editor

PLOS ONE

Additional Editor Comments (optional):

Dear Dr. Silva;

Thank you for submitting the revised version of your interesting work. I am glad to accept the manuscript at this time.

Reviewers' comments:

Reviewer's Responses to Questions

**Comments to the Author**

1. If the authors have adequately addressed your comments raised in a previous round of review and you feel that this manuscript is now acceptable for publication, you may indicate that here to bypass the “Comments to the Author” section, enter your conflict of interest statement in the “Confidential to Editor” section, and submit your "Accept" recommendation.

Reviewer #1: All comments have been addressed

Reviewer #2: All comments have been addressed

2. Is the manuscript technically sound, and do the data support the conclusions?

Reviewer #1: Yes

Reviewer #2: Yes

3. Has the statistical analysis been performed appropriately and rigorously? 

Reviewer #1: Yes

Reviewer #2: Yes

4. Have the authors made all data underlying the findings in their manuscript fully available?

Reviewer #1: Yes

Reviewer #2: Yes

5. Is the manuscript presented in an intelligible fashion and written in standard English?

Reviewer #1: Yes

Reviewer #2: Yes

6. Review Comments to the Author

Reviewer #1: (No Response)

Reviewer #2: Although the presentation of this version has been relatively confused displaying all the suggested changes, the authors have adequately addressed my comments and significantly improved the manuscript.

7. PLOS authors have the option to publish the peer review history of their article (what does this mean?). If published, this will include your full peer review and any attached files.

Reviewer #1: **Yes: **Mariana Balestrin

Reviewer #2: No

---

## [Editor Report · Acceptance letter]

16 Sep 2021

PONE-D-20-38212R1 

Time-trend in excess weight in Brazilian adults: a systematic review and meta-analysis 

Dear Dr. Silva:

I'm pleased to inform you that your manuscript has been deemed suitable for publication in PLOS ONE. Congratulations! Your manuscript is now with our production department. 

Kind regards, 

on behalf of

Dr. Cesario Bianchi 

Academic Editor

PLOS ONE